# PTRN-1 (CAMSAP) and NOCA-2 (NINEIN) are required for microtubule polarity in *Caenorhabditis elegans* dendrites

Liu He[1], Lotte van Beem[1], Berend Snel[2], Casper C. Hoogenraad[1,3], Martin Harterink[1]*

**1** Cell Biology, Department of Biology, Faculty of Science, Utrecht University, Utrecht, the Netherlands, **2** Theoretical Biology and Bioinformatics, Department of Biology, Faculty of Science, Utrecht University, Utrecht, the Netherlands, **3** Department of Neuroscience, Genentech, Inc., South San Francisco, California, United States of America

* m.harterink@uu.nl

**Data Availability Statement:** All relevant data are within the paper and its Supporting Information files.

## Abstract

The neuronal microtubule cytoskeleton is key to establish axon-dendrite polarity. Dendrites are characterized by the presence of minus-end out microtubules. However, the mechanisms that organize these microtubules with the correct orientation are still poorly understood. Using *Caenorhabditis elegans* as a model system for microtubule organization, we characterized the role of 2 microtubule minus-end related proteins in this process, the microtubule minus-end stabilizing protein calmodulin-regulated spectrin-associated *protein* (CAMSAP/PTRN-1), and the NINEIN homologue, NOCA-2 (noncentrosomal microtubule array). We found that CAMSAP and NINEIN function in parallel to mediate microtubule organization in dendrites. During dendrite outgrowth, RAB-11-positive vesicles localized to the dendrite tip to nucleate microtubules and function as a microtubule organizing center (MTOC). In the absence of either CAMSAP or NINEIN, we observed a low penetrance MTOC vesicles mislocalization to the cell body, and a nearly fully penetrant phenotype in double mutant animals. This suggests that both proteins are important for localizing the MTOC vesicles to the growing dendrite tip to organize microtubules minus-end out. Whereas NINEIN localizes to the MTOC vesicles where it is important for the recruitment of the microtubule nucleator γ-tubulin, CAMSAP localizes around the MTOC vesicles and is cotranslocated forward with the MTOC vesicles upon dendritic growth. Together, these results indicate that microtubule nucleation from the MTOC vesicles and microtubule stabilization are both important to localize the MTOC vesicles distally to organize dendritic microtubules minus-end out.

## Introduction

The microtubule cytoskeleton is vital in neurons for proper axonal and dendritic development. In axons, microtubules are mainly arranged with their plus-ends distal to the cell body. In contrast, dendritic microtubules in invertebrates are predominantly arranged with their minus-

**Funding:** This work was funded by the Nederlandse Organisatie voor Wetenschappelijk Onderzoek (NWO) (NWO-ALW-VICI 865.10.010 to C.C.H.), by the European Research Council (ERC Consolidator Grant 617050 to C.C.H.) and by the Chinese Scholarship Council (CSC) to L.H. The funders had no role in study design, data collection and analysis, decision to publish, or preparation of the manuscript.

**Competing interests:** The authors have declared that no competing interests exist.

**Abbreviations:** CAMSAP, calmodulin-regulated spectrin-associated protein; MAP, microtubule-associated protein; MTOC, microtubule-organizing center.

ends distal to the cell body or have a mixed orientation in vertebrates [1–3]. This difference in microtubule organization allows for selective cargo transport into axons or dendrites [4,5]. Defects in this organization can lead to problems in protein trafficking, neuronal development, and function [6–8]. Although the importance of the microtubule cytoskeleton organization is apparent, the molecular mechanism controlling differential microtubule organization between axons and dendrites is still not fully clear.

During cell division, the centrosome is the main microtubule-organizing center (MTOC). However, in polarized cells such as neurons, most microtubules are organized in a noncentrosomal manner [9–11]. Several mechanisms have been proposed to organize the neuronal microtubule cytoskeleton. These include the transport of microtubules into the correct organization (also referred to as microtubule sliding), microtubule growth from the cell body into the axon, the local nucleation of microtubules in the axon, and the selective stabilization of correctly organized microtubules by microtubule-associated proteins (MAPs) [5,12–14]. The formation of the axon is typically regarded as the initial step in neuronal polarization. Stabilization of axonal microtubules is a critical early event to form the axon. Indeed, the artificial stabilization of microtubules using drugs leads to the induction of multiple axons [15,16], and the formation of axons in cultured hippocampal neurons requires the TRIM46 protein that stabilizes and bundles the microtubules in a parallel plus-end out fashion in the proximal axon [17–19]. This plus-end out microtubule organization can be propagated into the growing axon by several mechanisms: by the outgrowth of existing microtubules potentially followed by severing [5,20], by forward translocation of the microtubule bundle [21], and by the local nucleation of new microtubules along the lattice of preexisting microtubules using the Augmin complex [22,23]. The transport of small microtubule fragments from the cell body has also been observed, but so far, it is not known if this contributes to the propagation of the axonal cytoskeleton [24].

While much is known about axonal microtubule organization, how dendrites acquire and maintain their typical minus-end out oriented microtubules is less clear. In mammals, Augmin-mediated microtubule nucleation and various MAPs were shown to contribute to the mixed microtubule organization [5,22,23,25]. In invertebrate neurons, the uniform minus-end out microtubule organization may offer a simpler starting point to understand the origin of the minus-end out microtubules. Early work suggested a role for microtubule sliding; in *Drosophila*, microtubules can be slid by the motor protein kinesin-1 during the earliest phases of neuron development [26], and in *C. elegans*, a mutant for kinesin-1 loses minus-end out microtubules in dendrites [27]. Recently, the role of local microtubule nucleation has gained attention, since noncentrosomal MTOCs are found localized in the dendrites and may be important to locally nucleate the minus-end out microtubules [14,28–30]. For example, Golgi outposts and early endosomes have been shown to nucleate microtubules in *Drosophila* dendrites, although the contribution to the overall microtubule organization is disputed [29,31–34]. In *C. elegans*, the relation between local microtubule nucleation and minus-end out microtubule organization in dendrites is clearer. It was shown that RAB-11-positive endosomes localize to the dendrite growth cone to nucleate microtubules with minus-end out organization [28]. However, it is unclear how the microtubule nucleating γ-tubulin is recruited to the RAB-11 vesicles and how specifically the minus-end out microtubules are maintained in dendrites.

A prominent protein family regulating microtubule stabilization is the CAMSAP family: CAMSAP1–CAMSAP3 (in vertebrates), Patronin (in *Drosophila*), and PTRN-1 (in *C. elegans*). These proteins can bind microtubule minus-ends and thereby protect them against depolymerization [35–38]. Indeed, in cultured mammalian neurons, microtubule stabilization by CAMSAP proteins was critical for neuronal polarization [25,39,40], and also in *Drosophila*,

Patronin was found to be important for dendritic microtubule polarity [41,42]. In addition to CAMSAP, microtubules can be stabilized by MAPs that can crosslink microtubules together or connect them to cortical structures via Ankyrin or Spectraplakin proteins [19,43]. In *C. elegans*, we found that the UNC-33(CRMP)/UNC-119/UNC-44(Ankyrin) complex connects the microtubule cytoskeleton to the cortex in both axons and dendrites to maintain the proper polarity organization [6].

In this study, we found that PTRN-1 (CAMSAP) is important in *C. elegans* for the proper localization of the MTOCs vesicles and, therefore, minus-end out microtubule polarity in the growing dendrite. Moreover, we found that the Ninein homologue NOCA-2 acts in parallel with PTRN-1 to localize γ-tubulin to the MTOC vesicles. Our results suggest that microtubule nucleation from the MTOC vesicles acts together with microtubule stabilization by CAMSAP proteins to organize dendritic microtubules minus-end out.

## Results

### Microtubule organization during early neuronal development depends on PTRN-1 (CAMSAP) and UNC-33 (CRMP)

The *C. elegans* PVD neuron is an excellent model to study neuronal development in vivo [44]. The microtubule cytoskeleton is for a large part restricted to the primary branches where it is organized with minus-end out polarity in the anterior dendrite, and with plus-end out polarity in the axon and posterior dendrite (Fig 1A) [7,45]. To address how these differences in microtubule organization are set up and maintained, we compared the contribution of 2 microtubule stabilizing proteins using genetic mutants: UNC-33 (CRMP) and PTRN-1 (CAMSAP). To visualize microtubule orientation, we used the microtubule plus-end protein EBP-2 fused to GFP or mKate2, which localizes to the microtubule plus-end only during microtubule growth. We found that both *unc-33* and *ptrn-1* mutations affect the microtubule organization in the anterior dendrite of the mature PVD neuron but in distinct manners (Fig 1B–1D). Whereas loss of *unc-33* led to a fully penetrant mixed or reversed microtubule polarity phenotype (as reported before [6,46]), the loss of *ptrn-1* led to occasional full reversal of microtubule polarity (Fig 1B–1D). This suggests that these proteins function differently in organizing neuronal microtubules. In the nonciliated PHC neuron and the ciliated URX neuron, we did not observe microtubule organization defects in the *ptrn-1* mutant (S1A and S1B Fig), indicating that the PTRN-1 requirement to organize the microtubules varies between neurons.

We previously found that the *unc-33* (CRMP) mutant phenotype can be rescued by reexpression of UNC-33L using the *des-2* promoter (Fig 1C) [6]. This promoter drives expression after the initial neurite outgrowth [46], suggesting that UNC-33 is important at later stages of neuronal development. In contrast, we found that the *ptrn-1* (CAMSAP) mutant phenotype was not rescued by expression of PTRN-1 using the *des-2* promoter, but it was rescued using the earlier expressing *unc-86* promoter (Fig 1D) [47]. This suggests that PTRN-1 functions earlier than UNC-33 to organize microtubules in the PVD neuron.

It was recently shown that during the outgrowth of the anterior PVD dendrite, distal RAB-11-positive vesicles function as an MTOC to nucleate microtubules towards the cell body to organize the dendritic microtubules minus-end out (Fig 1E) [28]. Indeed, when imaging EBP-2::GFP dynamics during early dendrite development, we saw pronounced microtubule growth events in the distal dendrite in both anterograde and retrograde directions (Fig 1F). This was not detected in mature neurons, suggesting that microtubule nucleation from MTOC vesicles mainly acts during development (S1C and S1D Fig). When analyzing microtubule dynamics in the *unc-33* mutant, we could still detect the pronounced distal microtubule dynamics in most animals during development (Fig 1G), even though mature neurons have a fully

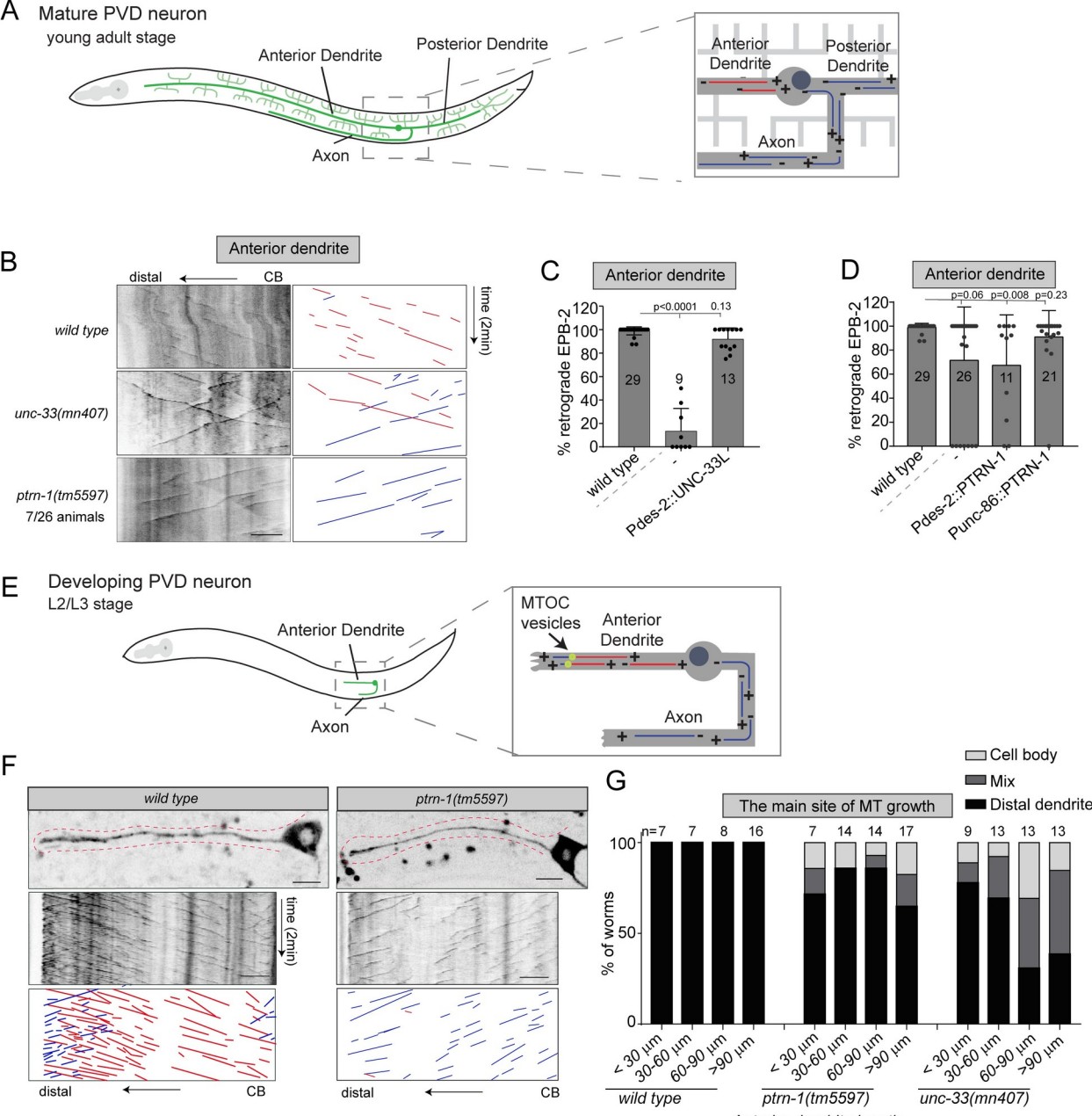

**Fig 1. Initial microtubule polarity establishment partially depends on PTRN-1 (CAMSAP) and UNC-33 (CRMP).** (**A**) Schematic representation of the microtubule organization in mature PVD neuron; the anterior dendrite contains uniform minus-end out microtubules (red), whereas the axon and posterior dendrites contain uniform plus-end out microtubules (blue) [7]. (**B-D**) Analysis of microtubule organization in the proximal PVD anterior dendrite using EBP-2::GFP to visualize the microtubule plus-end growth events. (**B**) Kymograph illustrating the EBP-2:: GFP growth events in the PVD anterior dendrite; 7 out of 26 animals have completely reversed microtubules orientation in *ptrn-1* mutant. (**C**) Quantification of the percentage of retrograde growth events in *unc-33* mutant with or without the *des-2* promoter-driven rescue construct (late expression). (**D**) Quantification of the percentage of retrograde growth events in *ptrn-1* mutant with or without the *des-2* (late expression) or *unc-86* promoter (early expression)-driven rescue constructs. Scale, 5 μm. Analyzed animals were from the L4 or young adult stage; error bars represent SD; statistical analysis, Kruskal–Wallis test followed by Dunn's multiple comparisons test. Number of analyzed animals is indicated. (**E**) Schematic representation of the microtubule organization in the developing PVD neuron. In the anterior dendrite, a distal MTOC localized in the anterior dendrite generates dendritic minus-end out microtubules and short plus-end out microtubules that grow towards the dendritic tip [28]. (**F, G**) Analysis of microtubule organization in the developing PVD anterior dendrite using EBP-2::GFP. Representative kymographs are shown (**F**), and the main site of microtubule growth events was visually classified (**G**) as mainly in the distal dendrite (distal), throughout the dendrite (mix) or as coming from the cell body (cell body). Scale, 5 μm; the number of analyzed animals is indicated; red line indicates the growing dendrite. The data underlying the graphs shown in the figure can be found in S1 Data.

penetrant phenotype (Fig 1C). We observed a gradual increase in the phenotype upon neuron maturation (Fig 1G), suggesting that UNC-33 is continuously required to organize the microtubules. Since UNC-33 (CRMP) forms a complex with UNC-44 (Ankyrin) and UNC-119 (UNC119) to anchor microtubules to the cortex [6] and that we found similar defects in the *unc-119* mutant (S1E Fig), this suggests that loss of cortical microtubule anchoring is the cause of this increasing phenotype. In the *ptrn-1* (CAMSAP) mutant, a small population of animals lost the pronounced distal microtubule growth events at early developmental stages and had reversed microtubule polarity organization (Fig 1F and 1G). In contrast to *unc-33* mutants, the fraction of animals with this defect remained constant during development and is similar to the defects seen in adult animals (Fig 1D). This suggest that PTRN-1 mainly acts at early stages to ensure distal microtubule nucleation to set up minus-end out microtubule polarity.

Together, these results indicate that the microtubule stabilizing proteins UNC-33 (CRMP) and PTRN-1 (CAMSAP) both act in the PVD neuron to regulate microtubule organization. However, they function differently: PTRN-1 mainly functions at early stages, whereas UNC-33 acts continuously during neuronal development to maintain microtubule polarity by connecting the microtubule cytoskeleton to the cortex.

## PTRN-1 (CAMSAP) and NOCA-2 (NINEIN) function in parallel to organize microtubules minus-end out in dendrites

Since loss of *ptrn-1* (CAMSAP) only shows a partially penetrant phenotype, this suggests that other factors work in parallel to set up dendritic microtubule organization in the PVD neuron. Previously, *ptrn-1* was shown to act in parallel to *noca-1* to organize the noncentrosomal microtubule organization in the *C. elegans* epidermis [48]. NOCA-1 has some sequence and functional similarity to mammalian NINEIN [48], but most likely is the *C. elegans* CEP85 homologue (S2A Fig). BLAST searches using mammalian NINEIN, identified another protein in *C. elegans*, which we called NOCA-2 (S2A and S2B Fig). Using CRISPR/Cas9, we generated the *noca-2(hrt28)* mutant, which has a 4,782-bps deletion encompassing the entire *noca-2* coding sequence (S2C Fig). We found that single mutants for either *noca-1* or *noca-2* showed a low penetrance microtubule polarity reversal phenotype in the mature PVD neuron, similar to the *ptrn-1* mutant (Figs 2A, 2B, S3A and S3B). This defect seems specific to the minus-end out microtubules, as the microtubule polarity in the PVD axon and posterior dendrites were unaffected (S3C–S3F Fig). Combining *ptrn-1* with either *noca-2* or *noca-1* led to a strong enhancement of the phenotype (Figs 2A, 2B, S3A and S3B). This suggest that *ptrn-1* acts in parallel to *noca-1* and *noca-2* in the PVD neuron, similar to what was observed before in the epidermis for *ptrn-1* and *noca-1* [48]. The *ptrn-1;noca-2* double mutant animals are superficially wild-type animals, while the *ptrn-1;noca-1* double mutant worms exhibit severe developmental defects (S3H Fig) [48]. Therefore, we cannot rule out that the neuronal defects are secondary to defects in the surrounding epidermis. Moreover, we were not able to rescue the *noca-1* mutant phenotype using 2 previously generated functional tagged transgenes (S3G Fig) [48]; therefore, we chose to here focus our studies on *noca-2*. Expression of NOCA-2 in the PVD neuron using the early expressing *unc-86* promoter was able to fully suppress the microtubule defect (Fig 2C), which may suggest that *noca-2* is also needed during early PVD development for proper microtubule organization. Indeed, looking at early dendrite development, we observed a low percentage of *noca-2* mutant animals that completely lost distal microtubule dynamics (Fig 2E). This phenotype that was strongly enhanced in the *ptrn-1;noca-2* double mutant (Fig 2D and 2E and S1 Video), similar to the mature neuron situation (Fig 2B). Quantification of the distal microtubule dynamics using EBP-2::GFP or GFP::TBA-1 (Tubulin) in

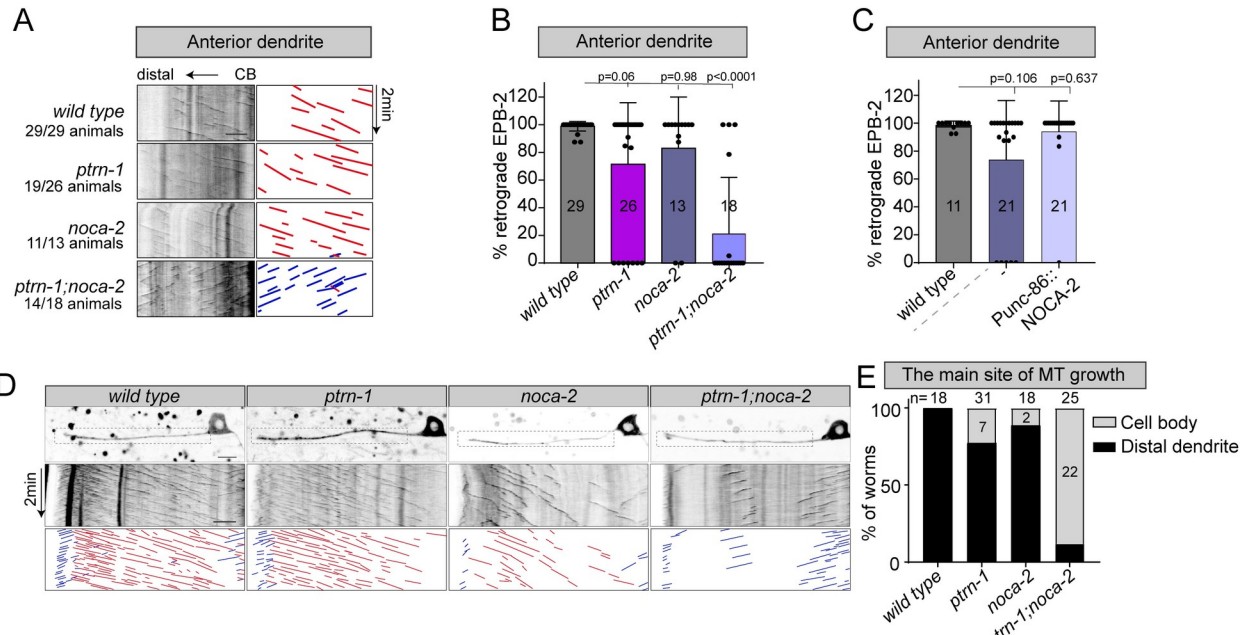

**Fig 2. PTRN-1 (CAMSAP) and NOCA-2 (NINEIN) act in parallel to organize microtubules minus-end out in dendrites.** (**A-C**) Analysis of microtubule organization in the mature PVD anterior dendrite using EBP-2::GFP to visualize the microtubule plus-end growth events. (**A**) Kymographs illustrating EBP-2::GFP growth events in the PVD anterior dendrite in the wild-type and indicated mutants. The numbers of animals with this phenotype are indicated. (**B**) Quantification of the retrograde growth events in mature PVD anterior dendrite in the wild-type and indicated mutant backgrounds. (**C**) Quantification of the percentage of EBP-2 retrograde growth events in *noca-2* mutant with or without the *unc-86* promoter (early expression)-driven rescue construct. Scale, 5 μm. Analyzed animals were from the L4 or young adult stage; error bars represent SD; statistical analysis, Kruskal–Wallis test followed by Dunn's multiple comparisons test. Number of analyzed animals is indicated. (**D, E**) Analysis of microtubule organization in the developing PVD anterior dendrite using EBP-2::GFP in indicated mutants. Representative kymographs are shown (**D**), and the main site of microtubule growth events was visually classified (**E**) as mainly in the distal dendrite (distal dendrite) or as coming from the cell body (cell body). Scale, 5 μm. Error bars represent SD; statistical analysis, Kruskal–Wallis test followed by Dunn's multiple comparisons test. Number of analyzed animals is indicated. The data underlying the graphs shown in the figure can be found in S1 Data.

the animals that retained the distal dynamics showed reduced growth events mainly in the *noca-2* mutant background (S4A–S4D Fig and S1 Video). Similarly, the microtubule plus-end growth speed and distance decreased upon *noca-2* depletion, whereas *ptrn-1* had no or very subtle defects (S4A and S4B Fig and S1 Video). This suggest that, although the *noca-2* and *ptrn-1* single mutants have similar microtubule organization defects in mature neurons, these proteins act differently during development.

Since the microtubule cytoskeleton is important for development and growth of neurons [5,49], we analyzed the morphology of the mature PVD neuron. Although a fraction of the *ptrn-1* and *noca-2* single mutants have reversed microtubule polarity (Fig 1A and 1B), we did not detect developmental defect in the mature neuron (S5 Fig). This suggests that loss of microtubule polarity alone does not lead to major developmental defect in the PVD neuron. The *ptrn-1;noca-2* double mutant, however, displayed pronounced morphological defects; it had reduced dendritic complexity especially in the anterior segments (S5A and S5B Fig), and the primary anterior dendrite was shorter (S5C Fig). We did not observe obvious defects in the posterior dendrite or the axon (S5B and S5D Fig).

Taken together, these results show that PTRN-1 (CAMSAP) and NOCA-2 (NINEIN) act in parallel in the PVD neuron to establish minus-end out microtubule organization and for proper dendritic morphogenesis.

## PTRN-1 (CAMSAP) and NOCA-2 (NINEIN) are essential for MTOC vesicles transport during dendrite outgrowth

It was previously shown that the microtubule nucleator γ-tubulin is transported on RAB-11-positive endosomes to the tip of the developing dendrite in the PVD neuron to establish the minus-end out microtubules [28]. Indeed, we detected dynamic GIP-2::GFP puncta (γ-tubulin small complex subunit 2) [48] in the distal segment of the anterior dendrite (Figs 3A and S8A), and these puncta perfectly overlapped with the RAB-11 marker (Fig 3B and 3C). To further investigate whether the microtubule polarity defects we observed in the *ptrn-1* (CAMSAP) and *noca-2* (NINEIN) mutants were caused by mislocalization of RAB-11 endosomes, we performed time-lapse imaging of PVD neurons expressing GFP::RAB-11 during neuron development. In wild-type animals, RAB-11 vesicles were consistently presents at the tip of the growing dendrite (Fig 3D and 3F and S2 Video). In the *ptrn-1* mutant, some animals lost distal RAB-11 vesicles, although most animals (13 out of 15) kept RAB-11 vesicles localized in the growing anterior dendrite (Fig 3D and 3F). Similarly, in absence of NOCA-2, 24 out of 29 animals had RAB-11 vesicles distally localized in anterior dendrite (Fig 3D and 3F and S2 Video). In contrast to the single mutants, in the double mutant, RAB-11 vesicles were almost entirely lost from the distal dendrite and instead localized to the cell body (Fig 3D and 3F and S2 Video). These defects closely resemble the microtubule defects we have observed in the mutants and strongly suggest that the loss of minus-end out microtubules in the dendrites of *ptrn-1* and *noca-2* mutants is caused by a loss of RAB-11 localization to the distal dendrite during outgrowth.

We also analyzed γ-tubulin localization using a GIP-2::GFP knock-in line [48]. As it is diffusely expressed in most tissues, only places with clear enrichment can be detected above the background signal. In wild-type animals, we consistently observed GIP-2 puncta in the distal dendrite (24/24 animals). In the *ptrn-1* (CAMSAP) mutant, GIP-2 puncta mislocalized to the cell body in 5 out of 23 animals (Figs 3E, 3G and S6A), which is consistent with the RAB-11 localization defects (Fig 3F) and microtubule polarity defect (Fig 1D and 1G). In the *noca-2* (NINEIN) mutant, however, we could not detect obvious GIP-2 clusters at the dendrite tip nor in the cell body in 12 out of 32 animals (Figs 3G and S6A). This suggests that γ-tubulin fails to be efficiently recruited to RAB-11 vesicles to sufficient levels that can be detected above background levels. When analyzing RAB-11 colocalization with GIP-2 in the *ptrn-1* and *noca-2* mutants, we indeed found that in the *noca-2* mutant, 5 out of 17 animals had a distal RAB-11 cluster without obvious GIP-2 enrichment, whereas in the *ptrn-1* mutant, all distal RAB-11 vesicles colocalized with GIP-2 (S6B and S6C Fig). Not surprisingly, *noca-2* mutants have a reduced number of GIP-2 puncta in the distal dendrite (Fig 3H) and reduced overall GIP-2 levels (Fig 3I), which agrees with the fact that specifically in *noca-2* mutants, we observed reduced microtubules dynamics in the distal dendrite (S4A–S4D Fig).

In conclusion, these results suggest that *noca-2* (NINEIN) works in parallel to *ptrn-1* (CAMSAP) to mediate distal MTOC localization to the growing dendrite but that the proteins act differently. Whereas CAMSAP proteins are well-described microtubule minus-end binding and stabilizing proteins that may stabilize tracks for MTOC vesicle transport, the function of NOCA-2 seems related to the efficient recruitment of γ-tubulin to the MTOC vesicles.

## NOCA-2 (NINEIN) acts at the MTOC vesicles

In order to visualize NOCA-2 (NINEIN) protein, we used CRISPR-Cas9 to endogenously insert a GFP tag at the C-terminus of the NOCA-2 locus. We observed NOCA-2 accumulations in several spots in the head and tail of the animal (Fig 4A) and at the cortex of the epidermal seam cells (Fig 4B). Indeed, using the NOCA-2 promoter to express mKate2, we observed a broad expression pattern (S6D Fig). Importantly, we detected NOCA-2::GFP in the distal

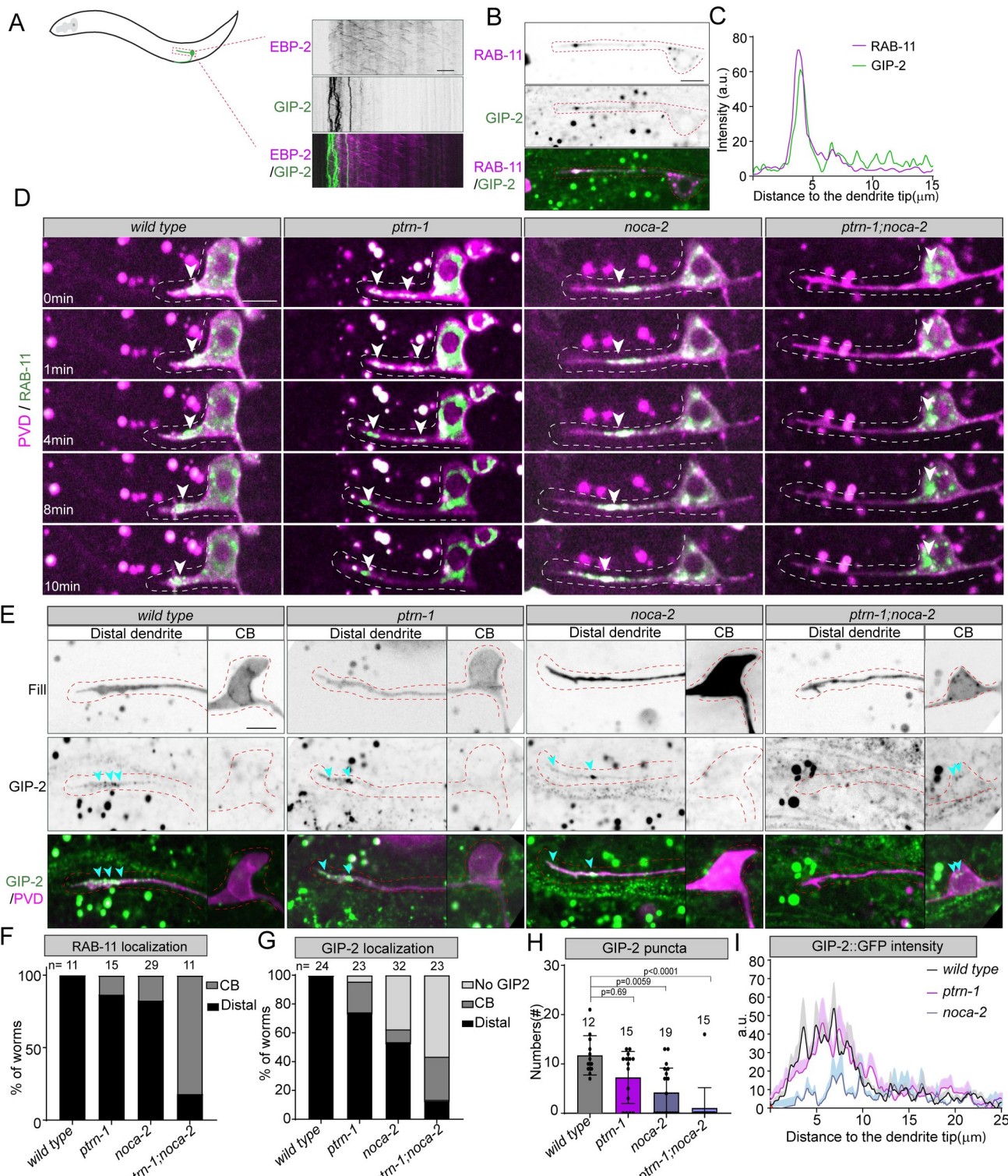

**Fig 3. PTRN-1 (CAMSAP) and NOCA-2 (NINEIN) are essential for distal MTOC vesicle localization during neurites outgrowth.** (**A**) Kymographs of the growing PVD anterior dendrite expressing EBP-2::mKate2 (PVD specific) and GIP-2::GFP (knock-in). Scale, 5 μm. (**B**, **C**) Colocalization of GIP-2:: GFP (knock-in) and mKate2::RAB-11 (PVD expressed) in the distal segment of the growing PVD anterior dendrite. Line scans for intensity profile of each channel from distal dendrites to cell body is shown in (**C**). Scale, 5 μm. PVD neurons are indicated with red dashed lines in (**B**). (**D**) Vesicle dynamics of PVD neurons expressing GFP::RAB-11 in the distal segment of the growing anterior dendrite in wild-type and indicated mutants. White arrowheads point to the dynamic RAB-11 clusters. Scale, 5 μm. Dashed line marks the developing dendrite. (**E**) GIP-2::GFP (knock-in) localization in the

wild-type and indicated mutants; green: GIP-2, magenta: fill of PVD neurons. GIP-2 cluster are indicated with blue arrowheads, and the PVD neurons are indicated with red dashed lines. Scale, 5 μm. (**F**, **G**) Quantification of RAB-11 (**D**) and GIP-2 (**E**) localization in wild-type and indicated mutants in the developing PVD neuron; light gray: no GIP-2 cluster was observed, dark gray: RAB-11 or GIP2 cluster localized in cell body, black: RAB-11 or GIP-2 accumulated in the distal developing dendrites. Number of analyzed animals is indicated. (**H**, **I**) Quantification of number of GIP-2::GFP puncta (**H**) and line scans of the average intensity from dendritic tip to cell body in the distal segment of the growing PVD anterior dendrite (**I**).Error bars represent SD; statistical analysis, Kruskal–Wallis test followed by Dunn's multiple comparisons test. Number of analyzed animals is indicated. The data underlying the graphs shown in the figure can be found in S1 Data.

PVD dendrite during neuronal development, which showed a similar localization pattern and dynamics as GIP-2 and RAB-11 (Fig 4C and 4D and S3 Video). In order to validate that NOCA-2 localizes to the MTOC vesicles, we expressed mKate2::RAB-11 in the PVD and observed clear colocalization (Figs 4E, 4F and S7A and S5 Video). In the mature neuron, we could not detect obvious NOCA-2 (nor GIP-2) accumulations at the dendrite tip (S7B and S7C Fig), which agrees with the loss of distal EBP-2 dynamics (S1C and S1D Fig). Since in the *noca-2* mutant γ-tubulin recruitment to the MTOC vesicles is reduced, this suggests that NOCA-2 localizes to the MTOC vesicles to recruit γ-tubulin for proper microtubule nucleation to organize dendritic microtubules minus-end out.

Mammalian NINEIN was shown to interact with γ-tubulin via its N-terminus [50]. Since this region is homologous to NOCA-2 (S2A Fig), it seems likely that the *C. elegans* proteins also interact. Validating this interaction by performing immunoprecipitations from endogenously tagged *C. elegans* did not work due to the very low expression levels of both proteins. Therefore, we performed immunoprecipitation experiments in HEK cells. However, using NOCA-2 as bait, we were not able to pull down mammalian γ-tubulin (S7D and S7E Fig). In addition, in the *C. elegans* seam cells, endogenous NOCA-2 only partially colocalizes with γ-tubulin (GIP-1) (S6E and S6F Fig), and the γ-tubulin localization is not obviously changed by artificial mislocalization of NOCA-2 to the mitochondria membrane (S6G Fig) or by mutation of *noca-2* (S6H Fig). These results argue against a direct interaction between NOCA-2 and γ-tubulin or that the interaction is regulated in a tissue-specific manner.

To further characterize the MTOC vesicles and better understand how NOCA-2 (NINEIN) and γ-tubulin may localize there, we have looked at 2 centrosomal proteins using GFP knock-in lines: the PCM scaffold protein SPD-5 and microtubule regulator TAC-1 (TACC) [51,52]. However, neither of these proteins localized to the MTOC vesicles in the PVD neuron (S8B Fig). This is consistent with a previous study that shows that SPD-5 mainly functions at the centrosomal MTOC [51,53,54]. Alternatively, NOCA-2 localization to the MTOC vesicles may be mediated by dynein. NINEIN proteins have been described as adaptors for dynein function [55,56], and in the *C. elegans* PVD, dynein was shown to localize to the MTOC vesicles to cluster these together [28]. To investigate whether NOCA-2 functions with Dynein, we measured the width of GIP-2 cluster localized in the distal PVD dendrite in *noca-2* and *ptrn-1* (CAMSAP) mutants. We found that the GIP-2 cluster in the *noca-2* mutant is similar to the *ptrn-1* mutant and only slightly wider than in wild-type animals (S8C Fig). In addition, distal DHC-1 (Dynein) accumulations showed no obvious difference between the *noca-2* mutant and the wild type (S8D Fig), arguing against a Dynein-mediated recruitment of NOCA-2 to MTOC vesicles. Therefore, more work is needed to understand how the RAB-11-positive vesicles in the PVD dendrite recruit the microtubule nucleation machinery.

## PTRN-1 (CAMSAP) puncta localize around the MTOC vesicles and translocate forward together

The involvement of the microtubule minus-end binding protein PTRN-1 (CAMSAP) in localizing the MTOC vesicles to the dendrite tip may suggest that PTRN-1 stabilizes a pool of

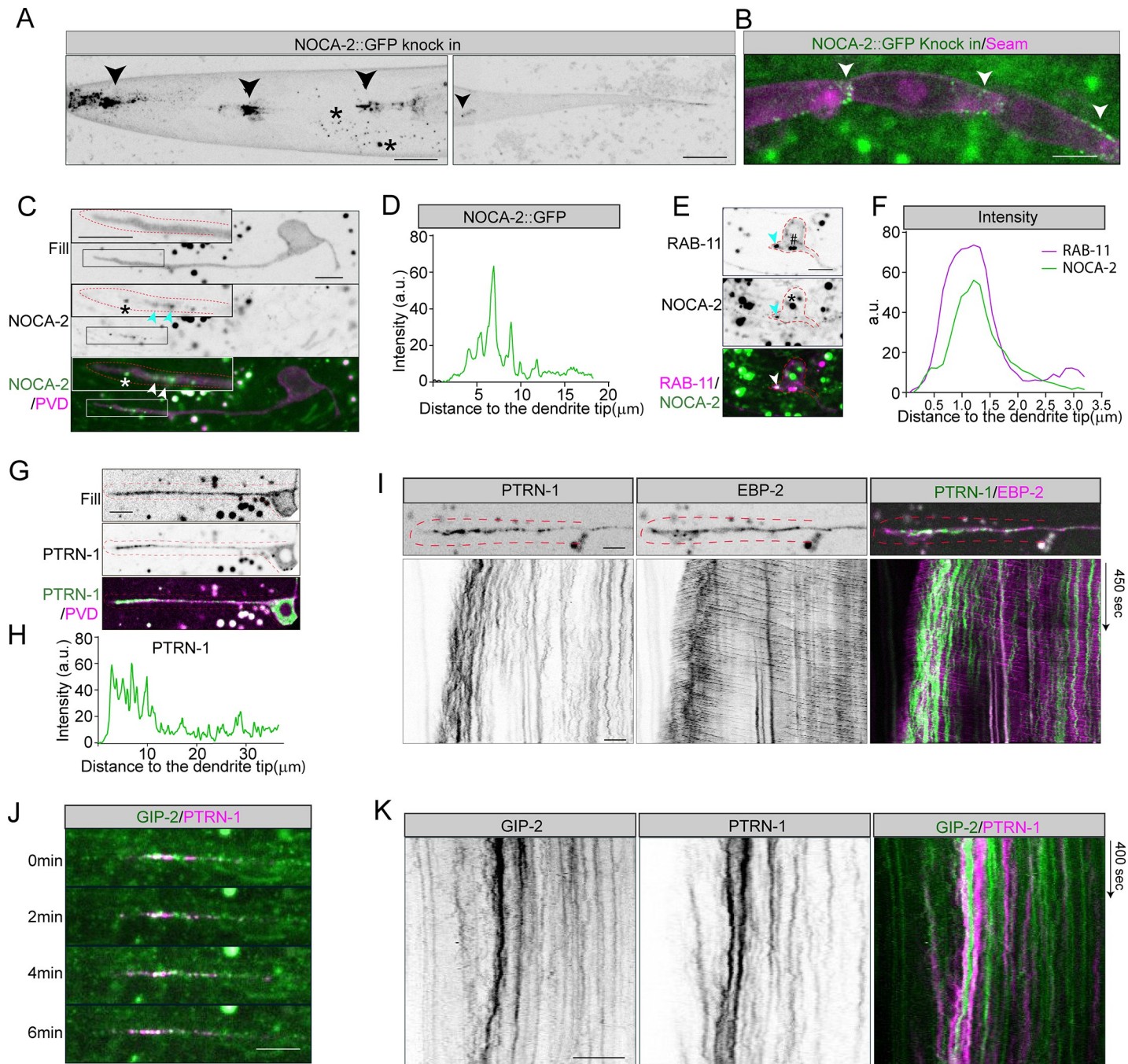

**Fig 4. NOCA-2 (NINEIN) localizes to the distal MTOC vesicles.** (**A**, **B**) The localization of endogenously tagged NOCA-2::GFP. NOCA-2 accumulates to structures in the head and tail (marked with arrowheads) (**A**) and to epidermal seam cell junctions (indicated with arrowheads) (**B**). The puncta marked with the * symbol are autofluorescent gut-granules. Scale, 10 μm. (**C**, **D**) Endogenous NOCA-2::GFP localization (**C**) and the intensity quantification (**D**) in the PVD neuron during anterior dendrite outgrowth (magenta: neuron fill). NOCA-2 puncta are indicated with arrowheads in (**C**) and shown enlarged in the top left of each panel (the puncta marked with * might be autofluorescent gut-granules). Scale, 5 μm. The developing anterior PVD dendrites are indicated with red dashed lines. (**E**, **F**) Colocalization (**E**) and the intensity profile (**F**) of endogenous NOCA-2::GFP (green) and PVD expressed mKate2::RAB-11 (magenta) in the developing PVD neuron. Arrowheads point to the colocalization of RAB-11 and NOCA-2. The puncta marked with * are autofluorescent, and the RAB-11 puncta without obvious NOCA-2 accumulation are marked with # in (**E**). Scale, 5 μm. The PVD neuron is indicated with red dashed lines. (**G**, **H**) Representative example (**G**) and intensity profile (**H**) of a PVD neuron expressing GFP::PTRN-1 in the developing dendrite; Fill of PVD (magenta), PTRN-1 puncta (green). Scale, 5 μm. The anterior PVD dendrite is indicated with red dashed lines. (**I**) PTRN-1 dynamics in the growing PVD anterior dendrites with respect to microtubule nucleation; PTRN-1(green), EBP-2::GFP (magenta). Scale, 5 μm. The distal anterior PVD dendrites are indicated with red dashed lines. (**J**, **K**) Cotransport and kymograph of GIP-2::GFP and mKate2::PTRN-1 in the distal segment of the growing PVD anterior dendrite. PTRN-1 (magenta), GIP-2 (green). Scale, 5 μm.

microtubules in the PVD neuron that are important for the forward translocation of the MTOC vesicles. Interestingly, it was proposed that a pool of transiently stabilized plus-end out microtubules in the tip of the PVD dendrites serve as tracks for the kinesin-1 motor (UNC-116) to move the MTOC vesicles forward [28]. To see if PTRN-1 could be involved in this process, we expressed GFP or mKate2-tagged PTRN-1 in the PVD neuron. We observed that PTRN-1 puncta distributed throughout the dendrites with a clear enrichment in the distal dendrite around the MTOC vesicles with no obvious colocalization (Fig 4G–4K). This distal accumulation was not observed mature dendrites (S7B and S7C Fig). To see if MTOC vesicles may be transported over the PTRN-1-decorated microtubules, we performed live imaging of PTRN-1. As expected, the PTRN-1 puncta in the mid-dendrite are largely immobile, suggesting that the microtubules are highly immobilized in the shaft (Fig 4I), e.g., by anchoring these to the cortex [6]. We did see occasional (weak) anterograde moving PTRN-1 puncta (S9D Fig), which could represent minus-end growth [41,57]. In the distal segment, however, the PTRN-1 puncta were more dynamic, suggesting a less anchored microtube cytoskeleton. More strikingly, when we imaged PTRN-1 together with the MTOC vesicle marker (GIP-2), we observed that their movement in the distal segments was correlated and that they translocated forward together upon dendrite growths (Fig 4J and 4K and S4 Video). To see if the microtubule cytoskeleton is translocating forward in the distal dendrite, we analyzed a marker for stable microtubules (UNC-116(Rigor)::GFP) [58]. The distal dendrite was largely devoid of the stable microtubule marker and only occasionally shows some microtubule dynamics (S9A and S9B Fig). This is consistent with the high EBP-2 and tubulin dynamics we observed in the distal segment (Figs 1F and S3C and S3D) and suggests a mainly dynamic pool of microtubules. We did not manage to track a marker for dynamic microtubules over time (UNC-104(Rigor):: GFP), probably reflecting the high microtubule turnover. Altogether, these results argue against kinesin-1 transport over CAMSAP-stabilized microtubules and instead suggest a connection between the MTOC vesicles and the CAMSAP puncta. More work is needed to understand the mechanism that transports the MTOC vesicles forward.

## Discussion

The presence of minus-end out microtubules in dendrites is one of the characteristic features that distinguish them from axons, allowing for selective cargo transport to set up neuronal polarity. It was recently shown in *C. elegans* that microtubule nucleation from MTOC vesicles in the distal dendrite is essential to organize dendritic microtubules minus-end out [28]. However, how these vesicles localize to the growing dendrite tip is incompletely understood, especially since these MTOC vesicles were suggested to be transported over the same microtubules as that they nucleate. This may suggest that these processes are connected and suggests a prominent role for selective microtubule stabilization. Here we report that the localization of these MTOC vesicles depends on the microtubule minus-end stabilizing protein PTRN-1 (CAMSAP), which functions in parallel to NOCA-2 (NINEIN) to set up dendritic minus-end out microtubules. Codepletion of PTRN-1 with NOCA-2 leads to MTOC vesicles mislocalization to the cell body, which is the underlying cause of the microtubule polarity defects.

Since these proteins function (partially) redundantly, this suggests that they act in different processes. Indeed, mutants for these genes have a different effect on microtubule properties in the distal dendrite and have a different localization pattern. PTRN-1 (CAMSAP) shows a punctate pattern throughout the dendrite, as expected for a protein that binds to microtubule minus-ends. We did see a clear enrichment of PTRN-1 puncta surrounding the MTOC vesicles, suggesting that there is a higher microtubule density in the distal segment (Fig 4I). NOCA-2 (NINEIN), on the other hand, localizes to the MTOC vesicles where it perfectly

overlaps with the microtubule nucleator γ-tubulin (Fig 4E and 4F). Moreover, in the *noca-2* mutant, we observed that γ-tubulin is not efficiently recruited to the MTOC vesicles (Figs 3G, S6B, and S6C), suggesting that NOCA-2 is involved in the recruitment of γ-tubulin for proper microtubule nucleation. However how NOCA-2 localizes to the vesicles and how it may recruit γ-tubulin is still an open question. Although the MTOC vesicles are RAB-11 positive, many RAB-11-positive endosomes localize to the cell body without obvious NOCA-2 accumulation (Fig 4E). Moreover RAB-11 depletion had only mild defects on GIP-2 localization [28]; therefore, it seems unlikely that RAB-11 directly recruits NOCA-2 and suggests that other proteins are involved. The endosomal recruitment of NOCA-2 may be aided by palmitoylation, as was previously shown for NOCA-1 [48]; http://lipid.biocuckoo.org/ predicts a high threshold palmitoylation site at C441 in NOCA-2. Alternatively, as human NINEIN can act as a dynein activator [55], NOCA-2 might act with dynein to localize to the MTOC vesicles [28]. However, we did not observe defects in MTOC vesicle clustering in the *noca-2* mutant as was reported for the dynein mutant [28], nor did we see obvious changes in dynein recruitment to the vesicles (S8D Fig).

In mammals, CAMSAP1, CAMSAP2, and CAMSAP3 all recognize and protect microtubules minus-ends against depolymerization and are important for neuron polarization [25,35,39,40]. However, their behavior at the minus-end is different: CAMSAP1 concentrates at the outermost ends and tracks the growing microtubule minus-ends, while CAMSAP2 and CAMSAP3 are stably deposited on the microtubule lattice, forming stretches from the minus-end and stabilizing MT lattices against depolymerization [35,59]. In *Drosophila* neurons, Patronin (CAMSAP) tracks and controls the microtubule minus-end, similar to CAMSAP1, and is important to populate dendrites with minus-end out microtubules [41]. In *C. elegans* PLM neurons, microtubule minus-ends growth was also reported in the posterior process [57]. Here, we found that most of GFP::PTRN-1 (CAMSAP) puncta were highly immobile in mature PVD neurons (S9D Fig). However, we did observe a small population of anterograde moving PTRN-1 puncta in the shaft of the anterior dendrite (S9D Fig). The speed of these movements was slower than the typical plus-end growth speeds (S9E Fig) and may represent microtubule minus-end growth, although these did not overlap with EBP-2::GFP as was the case in *Drosophila* (S9C Fig) [41]. Therefore PTRN-1 may contribute to the microtubule organization by stabilizing the microtubules in the dendrite shaft and potentially promoting minus-end growth to allow for the MTOC vesicle transport to the tip. In addition, we observed an enrichment of PTRN-1 puncta surrounding the MTOC vesicles that displayed very different dynamics, much slower and less processive (Fig 4I). Therefore, we do not expect the distal CAMSAP dynamics to represent microtubule growth. One potential model is that PTRN-1 (CAMSAP) stabilizes a specific subset of distal microtubules that are then used for forward translocation of the MTOC vesicles, e.g., by the UNC-116 (kinesin-1) motor over the short plus-end out microtubules in the tip [28]. However, we found that PTRN-1 localizes in a punctate pattern surrounding the MTOC vesicles and comigrate with the vesicles upon dendrite growth. This may suggest that microtubule nucleation is connected to microtubule minus-end stabilization and suggests to also consider alternative models where, e.g., pushing or pulling on microtubules may translocate the MTOC vesicles forward. Such forces could, for example, be generated by microtubule sliding over other microtubules. In *Drosophila*, kinesin-1 was shown to slide microtubule against other microtubules during early neuronal development, using an extra microtubule binding site in the tail [26,60,61]. Although kinesin-1 is also essential in *C. elegans* to organize microtubules minus-end out [27], mutating the extra microtubule binding site in the motor tail using CRISPR did no show microtubule defects arguing against a microtubule-microtubule sliding model for kinesin-1 in *C. elegans* (11 out of 11 animals had fully minus-end out microtubule organization). Alternatively, motors could push or pull the

microtubule cytoskeleton forward if anchored to static structures. For example, in axons, the dynein motor was shown to push the distal microtubule cytoskeleton forward by anchoring to the cortex and walking to the minus-end of the microtubules [21]. Similarly, kinesin-1 may push minus-end out–oriented microtubules towards the dendrite tip by walking to the plus-end. Or alternatively, dynein may function at the growing dendrite tip to pull on the short plus-end out microtubules that emanate from the MTOC vesicles, similar to its role to position the centrosomes during cell division [62–66]. Interestingly, the forward movements of the MTOC vesicles coincided with longer lived microtubules [28], which could represent cortically captured microtubules by dynein at the tip. More work is needed to determine the mechanism how the MTOC vesicles are transported anterogradely. Also, how the MTOC vesicles are connected to PTRN-1 stabilized microtubules will be interesting to investigate further. Potentially, NOCA-2 (NINEIN) can directly connect microtubules to the MTOC vesicles as *Drosophila* NINEIN (Bsg25D) was reported to bind to microtubules [67].

We propose that the minus-end out microtubule organization in the PVD dendrite follows a two-step model where non-centrosomal microtubules are initially nucleated from MTOC localized γ-tubulin and subsequently stabilized by PTRN-1 (CAMSAP). The minus-end out microtubule population may be further stabilized by cortical anchoring [6] and potentially further amplified by severing proteins such as Katanin and Spastin [68]. Such a mechanism may also take place in other tissues such as the *C. elegans* epidermis, where PTRN-1 and NOCA-1 (NINEIN) were found to also act in parallel to organize the noncentrosomal microtubules [48]. In contrast, in the *Drosophila* fat body, both Patronin (CAMSAP) and NINEIN act independent of γ-Tubulin to assemble noncentrosomal microtubules [69]. This indicates that the functional connection between microtubules nucleation and stabilization may vary between cell types and organisms.

## Materials and methods

### *C. elegans* strains and culturing

Strains were cultured at 15˚C or 20˚C using OP50 *Escherichia coli* as a food source and imaged at room temperature. To image early developing PVD neurons, the adult animals were grown at 15˚C for at least 48 h, and L2-L3 stage progeny were picked for imaging. The *noca-2(hrt28)* allele used in this study was made using CRISPR/Cas9-mediated mutagenesis, which deletes the entire NOCA-2 fragment (S1C Fig). The kinesin-1 allele with the mutated microtubule binding site in the tail (aa761RKKYQQ->AAAYAA) [27] was ordered from SunyBiotech and is called PHX1768 *unc-116(syb1768)*. *noca-2(hrt31[GFP])* was obtained by using CRISPR/Cas9-based genome editing [70]. The strains TV21539[tba-1(ok1135) I; wyEx8784[Punc-86:: gfp::tba-1; Punc-86:mCherry::PLCdeltaPH] and TV25056[*wyEx9975[Punc-86::gfp::rab-11.1 cDNA; Punc-86::mCherry::PLCdeltaPH]*] were gifts from Dr. Kang Shen [28]. The strain *gip-1 (wow25[tagRFP-t::3xMyc::gip-1]) III* was a gift from Dr. Jessica L. Feldman [71]. The strain *ntuIs6[Pdes-2::unc-116(G237A)::mCherry; Pdes-2::unc-104(E250K)::gfp; Podr-1::GFP]III* was a gift from Dr. Chan-Yen Ou [58]. The plasmid *Pdes-2::unc-116(G237A)::gfp* was cloned in *Pdes-2*::UNC-116(G237A)::mCherry (pCH9) that was provided by Dr. Chan-Yen Ou [58]. All trains used in this study are listed in S1 Table.

### DNA plasmids and gRNA

The DNA plasmids that were used to generate transgenic *C. elegans* strains and the detailed cloning information are listed in S1 Table. The gRNA used to make *noca-2* mutant and GFP knock-in stains are listed in S1 Table. *Pmyo2::mcherry* (5 ng/μl) was used as coinjection marker to generate extrachromosomal strains.

## Microscopy

For all imaging, *C. elegans* were mounted on 5% agarose pads with 10 mM tetramisole or 5 mM Levamisole solution in M9 buffer. Live imaging was performed within 60 min after mounting on a Nikon Eclipse-Ti microscope and with a Plan Apo VC, 60×, 1.40 NA oil or a Plan Apo VC 100 × N.A. 1.40 oil objectives (Nikon). The microscope is equipped with a motorized stage (ASI; PZ-2000), a Perfect Focus System (Nikon), ILas system (Roper Scientific France/PICT-IBiSA, Curie Institute) and uses MetaMorph 7.8.0.0 software (Molecular Devices) to control the camera and all motorized parts. Confocal excitation and detection are achieved using 100-mW Vortran Stradus 405 nm, 100-mW Cobolt Calypso 491 nm, and 100-mW Cobolt Jive 561 nm lasers and a Yokogawa spinning disk confocal scanning unit (CSU-X1-A1N-E; Roper Scientific) equipped with a triple-band dichroic mirror (z405/488/568trans-pc; Chroma) and a filter wheel (CSUX1-FW-06P-01; Roper Scientific) containing BFP (ET-DAPI (49000), GFP (ET-GFP (49002)) and mCherry (ET-mCherry(49008)) emission filters (all Chroma). Confocal images were acquired with a QuantEM:512 SCEMCCD camera (Photometrics) at a final magnification of 110 nm (60× objective), 67 nm (100× objective) per pixel, including the additional 2.0× magnification introduced by an additional lens mounted between scanning unit and camera (Edmund Optics). All EBP-2::GFP and GFP::TBA-1 imaging was performed at 1 frame per second (fps)

For the analysis of PVD neuron morphology, images were acquired using an LSM700 (Zeiss) confocal with a 20× NA 0.8 dry objective using the 488 nm and 555 nm laser lines.

## Quantitative image analysis

Image processing and analysis was done using ImageJ (FIJI) to create kymographs or merged sum intensity images for the intensity profile measurement. Statistical analysis and graphs were made in GraphPad Prism software version 8.0.

To quantify EBP-2::GFP growth orientation, kymographs were made using Kymograph Builder plugin in ImageJ. The retrogradely and the anterogradely growing EBP-2::GFP were manually counted. To quantify the microtubule polarity in the mature PVD dendrite, imaging was performed in the proximal dendrite. For the developing PVD dendrite, the whole anterior PVD region was imaged. To quantify EBP-2::GFP growing speed and frequency (S4A and S4B Fig), only the distal region (20 μm) was quantified.

To quantify the main site of microtubule growth (Figs 1G, 2E and S2A), kymographs were made for the whole of anterior dendrite in the developing neurons. When EBP-2 comets mainly grew from the distal anterior dendrites, the animal was classified as "distal dendrite"; when the EBP-2 comets mainly grew from cell body, theses were classified as "cell body"; and when EBP-2 comets grew in both directions, these were classified as "Mix."

To quantify the PVD branch complexity, the entire PVD dendrite was divided into 4 segments: One posterior segment (−1) and the anterior segment was divided into 3 equal length anterior segments (+1, +2, and +3) (Fig 2F). The "branch complexity" index calculation was based on a previous study definition [45].

To measure the GIP-2 cluster intensity profiles (Fig 3I), cytosolic mKate2 was used to visualize the dendrite tip. We drew a 50 px-wide line from the tip of the anterior dendrite to measure the intensity profile of the GIP-2 cluster and the background intensity was subtracted from the region next to the neuron devoid of gut granule autofluorescence.

Sequence analysis and phylogenetic tree

To investigate the relation between NOCA-1, NOCA-2, and human NINEIN (UniProt #Q9Y2I6), we performed BLAST searches using HHpred (https://toolkit.tuebingen.mpg.de/tools/hhpred) [72,73]. To identify domain structures, we made use of UniProt annotations

supplemented with the Aphafold2 predicted 3D structure [74]. To generate the phylogenetic tree, UniProt (https://www.uniprot.org/) was used to search for the NINEIN homologous sequences from different species. The phylogenetic tree was generated using the full length proteins and the online version of MAFFT (http://mafft.cbrc.jp/alignment/server/) [75].

## Pull-down and western blot

HEK293T cells (authenticated and tested negative for mycoplasma) were transfected with NOCA-2::EGFP or cotransfected with BirA together with bio-mCherry::NOCA-2 and incubated at 37˚C for 24 h. Cells were harvested and washed 1× with ice-cold PBS and lysed with lysis buffer (100 mM Tris-HCl (pH 7.5), 150 mM NaCl, 1% Triton X-100, and 1× protease inhibitor cocktail). Cell lysates were centrifuged at 13,000 rpm for 10 min, and the supernatants were incubated with GFP-Trap magnetic beads (Chromotek) or Dynabeads M-280 (Invitrogen). Beads were preblocked in buffer containing 20 mM Tris (pH 7.5), 20% glycerol, 150 mM NaCl, and 10 μg chicken egg albumin for 30 min and then washed twice with buffer containing 20 mM Tris (pH 7.5), 150 mM NaCl, and 0.1% Triton X-100. After incubating for 1 h at 4˚C, the supernatant was collected (after pull-down) and the beads were washed 5 times with washing buffer. Samples were eluted with SDS/DTT sample buffer and boiled for subsequent western blot. For western blot, samples were loaded onto 8% SDS-PAGE gels and transferred to nitrocellulose membrane. Membranes were blocked with 2% bovine serum albumin (BSA) in PBS/0.05% Tween-20.

For NOCA-2::EGFP transfected cells samples, primary antibodies: anti-GFP (ab290, Abcam, rabbit) and anti-γ-Tubulin (T6557, Sigma, mouse) were diluted in blocking buffer and incubated with the membranes overnight at 4˚C. For Bio-mCherry::NOCA-2 transfected cells, primary antibodies: anti-mCherry (632543, Clontech, mouse) and anti-γ-Tubulin (T3559, Sigma, rabbit) were used. After primary antibody incubation, membranes were washed 3 times with PBS/0.05% Tween 20 and incubated with secondary IRDye 680LT anti-mouse or IRDye 800LT anti-rabbit antibodies for 45 min at room temperature. Membranes were then washed 3 times with PBS/0.05% Tween 20 and scanned on Odyssey Infrared Imaging system (LI-COR Biosciences).

## Supporting information

**S1 Fig.** (**A**, **B**) Quantification of the percentage of retrograde EBP-2::GFP growth events in the ciliated PHC dendrites (**A**) and the nonciliated URX dendrites (**B**) in wild-type and the *ptrn-1* mutant. Error bars represent SD; statistical analysis is followed by unpaired Student *t* test. Number of analyzed animals is indicated. (**C**, **D**) Examples of EBP-2::GFP dynamics in the mature PVD neuron imaged in the distal and proximal anterior dendrite (**C**) and the quantification of EBP-2 growth frequency (**D**). Error bars represent SD; statistical analysis was performed with an unpaired Student *t* test. (**E**) The quantification of the main site of microtubule growth during neuron developing in the *unc-119* mutant visually classified as mainly in the distal dendrite (distal dendrite), throughout the dendrite (mix) or as coming from the cell body (cell body). The data underlying the graphs shown in the figure can be found in S1 Data. (TIF)

**S2 Fig.** (**A**) Schematic representation of NOCA-1, NOCA-2, and human NINEIN and CEP85 based on UniProt annotations and Alphafold2 predicted 3D structure. NOCA-2 and NINEIN are unambiguous orthologs as evidenced by phylogenetics (**B**) and domain composition, but the evolutionary relation between NOCA-1, NOCA-2, and NINEIN is not so obvious despite initial reports that NOCA-1 and NINEIN are orthologs [48]. Alphafold2 predictions of

NOCA-1, as available at https://alphafold.ebi.ac.uk/entry/G5EEK3, reveal no EF-hands or other globular domains that would normally allow unambiguous establishment of homology with NINEIN. Instead, they reveal a large coiled coil and a C-terminal set of parallel interacting alpha helices (marked with *). A profile search of this C-terminal region of NOCA-1 versus a database of profiles of human proteins hits CEP85 as best hit with e-value 0.0085 using HHPRED at the MPI-Toolkit on July 4. Reciprocal searches of full-length CEP85 profiles versus profiles of all *C. elegans* proteins hit NOCA-1 profile with 9.9e-21. These profile searches thus reveal that NOCA-1 and CEP85 are bidirectional best hits. In addition, Alphafold2 predicted structures of CEP85 reveal a similar set of interacting alpha helices in their C-term as predicted for NOCA-1, which is absent from NOCA-2 and NINEIN. Finally, both CEP85 and NOCA-1 share a C-terminal CxxQ putative farnesylation motif, which is also absent from NOCA-2 and NINEIN. Based on the reciprocal best profile-profile hits, structural and motif similarities, it is likely that CEP85 and NOCA-1 are homologs and that NOCA-1 is not homologous to NINEIN nor to NOCA-2. (**B**) Phylogenetic tree of full-length NINEIN and NINEIN Like (Ninl) proteins from various species. (**C**) Gene structure of the *noca-2* gene and the *hrt28* deletion.
(TIF)

**S3 Fig.** (**A**-**F**) Representative kymographs and quantification of microtubule polarity in the mature PVD neurons in the indicated mutants using EBP-2::GFP. The percentage of retrograde growing events in the anterior dendrite (**A**, **B**), in the posterior dendrite (**C**, **D**), and in the axon (**E**, **F**). Scale, 5 μm. (**G**) Quantification microtubule polarity using EBP-2::GFP in the anterior dendrite of wild-type and *ptrn-1;noca-2* mutants with or without 2 tagged NOCA-1 rescue constructs [48]. (**H**-**J**) Quantification of the PVD morphology. (**H**) Representative examples of the PVD morphologies in the indicated mutants. Scale, 20 μm. Quantification of (**I**) the PVD dendritic branch complexity [45]; (**J**) the relative axon length in the ventral nerve cord; (**K**) the relative length of the anterior dendrite. For microtubule polarity analysis, the animals were from L4 to young adult stage. For PVD morphology analysis, only young adult stage animals were analyzed. Error bars represent SD; statistical analysis, Kruskal–Wallis test followed by Dunn's multiple. The data underlying the graphs shown in the figure can be found in S1 Data.
(TIF)

**S4 Fig.** (**A**, **B**) Quantification of retrograde (**A**) and anterograde (**B**) microtubule plus-end dynamics in the distal 20 μm of the anterior PVD dendrite during neuron development using the plus tip marker EBP-2::GFP. For *ptrn-1* and *noca-2* mutants, only animals that retained distal microtubule nucleation were quantified. For speed measurements, only growth events of >2 μm were considered. Scale, 5 μm. Error bars represent SD; statistical analysis, Kruskal–Wallis test followed by Dunn's multiple comparisons test. Number of analyzed animals is indicated. (**C**) Representative kymographs of GFP::TBA-1 in the distal anterior dendrite. Red lines, growing plus-end out MTs in distal region. Scale bar, 5 μm. The distal anterior PVD dendrites are indicated with dashed lines. (**D**) Quantification of plus-end out microtubule polymerization frequencies in the distal region of the growing anterior dendrite. Only animals that displayed distal microtubule nucleation were considered. Error bars represent SD; statistical analysis, Kruskal–Wallis test followed by Dunn's multiple comparisons test. Five animals for each group were analyzed. The data underlying the graphs shown in the figure can be found in S1 Data.
(TIF)

**S5 Fig. Quantification of PVD morphology in the indicated mutants.** (**A**) Representative images of the PVD morphologies. Note that this marker is also expressed in another neuron (FLP) located in the head (left of the +3 region) and also in the coelomocytes (marked with *),

which were used as injection marker. (**B**) Quantification of PVD dendritic branch complexity in the 4 PVD regions along the anteroposterior axis as indicated in (**A**), based on [45]. (**C**) Quantification of the anterior dendrite outgrowth towards the FLP cell body localized in the head. (**D**) Quantification of the relative axon length in the ventral nerve cord. Scale, 20 μm. Analyzed animals were young adult stage; Error bars represent SD; statistical analysis, Kruskal–Wallis test followed by Dunn's multiple comparisons test. The data underlying the graphs shown in the figure can be found in S1 Data.
(TIF)

**S6 Fig.** (**A**) Representative examples of endogenously tagged GIP-2::GFP in the PVD neuron. GIP-2 accumulated in the cell body in the *ptrn-1* mutant (second panel) and without obvious GIP-2 accumulation in *noca-2* mutant (third panels) and *ptrn-1;noca-2* mutants (last panels). Green: GIP-2, magenta: PVD neuron fill. GIP-2 puncta are indicated with blue arrowheads. Scale, 5 μm. The developing neurons are indicated with red dashed lines. (**B**) Example images of mKate2::RAB-11 (magenta) and GIP-2::GFP (green) colocalization in the distal segment of the growing PVD anterior dendrite. The localization of RAB-11 and GIP-2 in developing anterior dendrite is indicated with arrowheads. Scale, 5 μm. The PVD neuron is indicated with red dashed lines. (**C**) Quantification of the number of animals in which mKate2::RAB-11 colocalizes with GIP-2::GFP in the distal segment of the growing PVD anterior dendrite; gray: the percentage of animals that have RAB-11 accumulated in distal dendrites but without GIP-2 accumulation. Number of analyzed animals is indicated. (**D**) The expression pattern of mKate2 driven by 2 kb of the *noca-2* promoter sequence. Scale, 20 μm. (**E, F**) Examples of localization (**E**) and the intensity quantification (**F**) of endogenous NOCA-2 and GIP-1 in epidermal seam cells. Scale, 5 μm. The epidermal seam cells are marked with white lines. (**G**) Representative example images of the GIP-2 (green) localization upon artificial NOCA-2 (magenta) mislocalization to mitochondria by fusing it to TOMM-20 (1–41 amino acids). Scale, 5 μm. The epidermal seam cells are marked with white lines. (**H**) The localization of GIP-2 in epidermal seam cells in wild-type (upper panels) and in *noca-2* mutant (lower panels). Scale, 5 μm. The epidermal seam cells are marked with white lines. The data underlying the graphs shown in the figure can be found in S1 Data.
(TIF)

**S7 Fig.** (**A**) Examples of endogenous NOCA-2::GFP (green) and PVD expressed mKate2:: RAB-11 (magenta) at different developmental stages of PVD anterior dendrite. The colocalization is indicated with arrowheads in the merged image and the outline of the dendrite is marked by a red dashed line. Scale, 5 μm. (**B, C**) Representative example of endogenously tagged NOCA-2 and GIP-2 and PVD expressed PTRN-1 in the distal part of the mature PVD anterior dendrite (**B**) and a summary diagram of localization of localization of NOCA-2, GIP-2, and PTRN-1 in distal mature dendrites and developing dendrites (**C**). The distal anterior PVD dendrites are indicated by a dashed red line; * marks autofluorescent gut granules. Scale, 5 μm. (**D**) Pull-down of NOCA-2::EGFP and EGFP (control) from HEK293T cells using GFP-Trap magnetic beads. Anti-GFP and anti-γ-tubulin were used to detected NOCA-2 and GFP (left panel) and human γ-tubulin (right panel). (**E**) Streptavidin pull-down assays from HEK293T cells coexpressing with bio-mCherry-NOCA-2 or bio-mCherry with BirA. Anti-mCherry and anti-γ-tubulin were used to detected NOCA-2 or mCherry (left panel) and human γ-tubulin (right panel).
(TIF)

**S8 Fig.** (**A**) Stills and kymograph of endogenous GIP-2::GFP localization over time (left panel) in growing PVD anterior dendrite. Scale, 5 μm. The distal dendrite is indicated with red

dashed lines. (**B**) GFP::SPD-5 and GFP::TAC-1 localization at the growing PVD anterior dendrite using endogenously tagged strains. (**C**) Quantification of GIP-2 cluster width in the growing PVD anterior dendrite. (**D**) Multiple examples of endogenous DHC-1::GFP (green) and PVD expressed mKate2::RAB-11 (magenta) in the growing PVD anterior dendrite. Scale, 5 μm. The distal anterior PVD dendrites are indicated with white dashed lines. The data underlying the graphs shown in the figure can be found in S1 Data.
(TIF)

**S9 Fig.** (**A**, **B**) Stills (**A**) and kymographs (**B**) of UNC-116(rigor)::GFP localization in the PVD anterior dendrites (A, green). Myristoylated mKate2 was used as a fill (magenta). Scale, 5 μm. (**C**, **D**) Representative example kymograph of EBP-2::GFP (**C**) and mKate2::PTRN-1 (**D**) in the mature PVD anterior dendrite. Examples of moving PTRN-1 puncta are indicated with blue arrowheads. Scale, 5 μm. (**E**) Quantification of EBP-2::GFP growth speed and PTRN-1 moving speed in the mature PVD anterior dendrite. Error bars represent SD; statistical analysis is followed by unpaired Student *t* test. The data underlying the graphs shown in the figure can be found in S1 Data.
(TIF)

**S10 Fig.** (**A**, **B**) Model for NOCA-2 (NINEIN) and PTRN-1 (CAMSAP) functioning at the growing dendrite tip of the PVD neuron. NOCA-2 localizes to the MTOC endosomes localized to the dendrite tip and is involved in recruiting the microtubule nucleating γ-tubulin, whereas PTRN-1 localizes around the MTOC vesicles where it may stabilize the nucleated microtubules.
(TIF)

**S1 Video. EBP-2::GFP comets in the developing anterior PVD dendrite in wild-type and indicated mutants.** Time, min:sec.
(AVI)

**S2 Video. GFP::RAB-11 (green) cluster movement in growing PVD anterior dendrite (magenta) in wild-type and indicated mutants.** Time, min:sec.
(AVI)

**S3 Video. The accumulation and dynamic of NOCA-2::GFP in distal developing anterior dendrites.** Time, min:sec.
(AVI)

**S4 Video. Comovement of mKate2::PTRN-1 (magenta) and the MTOC marker (GIP-2:: GFP, green) in developing PVD anterior dendrite.** Time, min:sec.
(AVI)

**S5 Video. Colocalization of NOCA-2 and RAB-11 in the growing PVD anterior dendrite.** GFP::RAB-11 (magenta) and NOCA-2::GFP (green). Time, min:sec.
(AVI)

**S1 Table. Overview of all *C. elegans* strains, constructs, and oligos used in this study.**
(XLSX)

**S1 Data. Data underlying the graphs.**
(XLSX)

**S1 Raw Images. Uncropped western blots—Raw data for S7 Fig.**
(PDF)

## Acknowledgments

We thank Mike Boxem and Sander van den Heuvel for advice, *C. elegans* reagents, and infrastructure. We thank Jason Kroll and Amélie Freal for feedback on the manuscript and Bart de Haan for helping with cloning. We thank Kang Shen for helpful suggestions and sharing of strains. We thank Jessica Feldman, Chan-Yen Ou, and Alexander Dammerman for kind sharing of *C. elegans* strains and reagents. Some strains were provided by the CGC, which is funded by the NIH Office of Research Infrastructure Programs (P40 OD010440), and some by the National Bioresource Project. We thank WormBase for curating and making available data related to *C. elegans*.

## Author Contributions

**Conceptualization:** Liu He, Casper C. Hoogenraad, Martin Harterink.

**Data curation:** Liu He.

**Formal analysis:** Liu He.

**Funding acquisition:** Liu He, Casper C. Hoogenraad, Martin Harterink.

**Investigation:** Liu He, Lotte van Beem, Berend Snel.

**Methodology:** Liu He.

**Project administration:** Liu He, Martin Harterink.

**Supervision:** Liu He, Martin Harterink.

**Validation:** Liu He, Lotte van Beem.

**Visualization:** Liu He, Martin Harterink.

**Writing – original draft:** Liu He, Martin Harterink.

**Writing – review & editing:** Liu He, Casper C. Hoogenraad, Martin Harterink.

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
