## [Editor Report · Decision Letter 0]

19 Feb 2022

Dear Dr Harterink, 

Thank you for submitting your manuscript reviewed by Review Commons entitled "PTRN-1/CAMSAP and NOCA-2/NINEIN are required for microtubule polarity in Caenorhabditis elegans dendrites" for consideration as a Research Article by PLOS Biology.

Your manuscript has now been evaluated by the PLOS Biology editorial staff as well as by an academic editor with relevant expertise and I am writing to let you know that we would like to consider a revision of your submission.

However, in order to invite you to submit a revision, we need you to complete your submission by providing the metadata that is required for full assessment. To this end, please login to Editorial Manager where you will find the paper in the 'Submissions Needing Revisions' folder on your homepage. Please click 'Revise Submission' from the Action Links and complete all additional questions in the submission questionnaire.

Once your full submission is complete, your paper will undergo a series of checks in preparation for peer review. Once your manuscript has passed the checks we will invite you to send us a revision. To provide the metadata for your submission, please Login to Editorial Manager (https://www.editorialmanager.com/pbiology) within two working days, i.e. by Feb 22 2022 11:59PM.

Given the disruptions resulting from the ongoing COVID-19 pandemic, please expect some delays in the editorial process. We apologise in advance for any inconvenience caused and will do our best to minimize impact as far as possible.

Kind regards,

Ines

--

Ines Alvarez-Garcia, PhD

Senior Editor

PLOS Biology

---

## [Editor Report · Decision Letter 1]

22 Feb 2022

Dear Dr Harterink,

Thank you for submitting your manuscript entitled "PTRN-1/CAMSAP and NOCA-2/NINEIN are required for microtubule polarity in Caenorhabditis elegans dendrites" for consideration as a Research Article at PLOS Biology. Your manuscript has been evaluated by the PLOS Biology editors and an Academic Editor with relevant expertise and we have considered the reviews submitted by Review Commons.

In light of the reviews, we would welcome submission of a revised version that takes into account the reviewers' comments. We cannot make any decision about publication until we have seen the revised manuscript and your response to the reviewers' comments. Your revised manuscript is also likely to be sent for further evaluation by the reviewers.

We expect to receive your revised manuscript within 3 months. 

**IMPORTANT - SUBMITTING YOUR REVISION**

3. Resubmission Checklist

a) *PLOS Data Policy*

b) *Published Peer Review*

Sincerely,

Ines

--

Ines Alvarez-Garcia, PhD

Senior Editor

PLOS Biology

---

## [Decision Letter · Decision Letter 2]

24 Jun 2022

Dear Dr Harterink,

Thank you for your patience while we considered your revised manuscript entitled "PTRN-1/CAMSAP and NOCA-2/NINEIN are required for microtubule polarity in Caenorhabditis elegans dendrites" for consideration as a Research Article at PLOS Biology. Your revised study has now been evaluated by the PLOS Biology editors, the Academic Editor and the two original reviewers. 

The reviews are attached below. As you will see, the reviewers are mostly satisfied with the improvements you have done in the manuscript, however they have raised a few remaining points that will need to be addressed before we can consider the paper for publication. While Reviewer 1 only notes two minor issues, Reviewer 2 thinks that you should focus the manuscript more on Noca-2, as it is newly described, giving a better overview on how this gene fits into the family. 

In light of the reviews, we are pleased to offer you the opportunity to address the remaining points from the reviewers in a revision that we anticipate should not take you very long. We will then assess your revised manuscript and your response to the reviewers' comments with our Academic Editor aiming to avoid further rounds of peer-review, although might need to consult with the reviewers, depending on the nature of the revisions.

**IMPORTANT - SUBMITTING YOUR REVISION**

3. Resubmission Checklist

a) *Data not shown*

Please note that per journal policy, we do not allow the mention of "data not shown", "personal communication", "manuscript in preparation" or other references to data that is not publicly available or contained within this manuscript. Please either remove mention of these data or provide figures presenting the results and the data underlying the figure(s).

b) *Published Peer Review*

Sincerely,

Ines

--

Ines Alvarez-Garcia, PhD

Senior Editor

PLOS Biology

Reviewers' comments

Rev. 1:

The authors have addressed all the reviewer's concerns and i feel the paper is ready for publication, after just a couple of minor text changes (see below). I particularly like the fact the authors have included "negative" data, which is refreshing to see. I think the paper is nicely written and the results are clearly described without being over-hyped. The paper is addressing an important question and while there are still questions to answer I feel the authors have done enough for publication in PlosBiology.

Minor comments

1) In the intro, the authors could also state that plus end growth from the soma into the axon can also be a mechanism that contributes to plus-end-out polarity in axons and cite Mukherjee et al., 2020.

2) When the authors say: "For example, Golgi outposts and early endosomes have been shown to nucleate microtubules in Drosophila dendrites and were suggested to contribute to the minus-end out microtubule organization in these neurites (Mukherjee et al., 2020; Ori-McKenney et al., 2012; Weiner et al., 2020; Ye et al., 2007)". The Mukherjee 2020 paper did not actually show nucleation from Golgi outposts, but rather than most Golgi outposts lacked g-tubulin. Instead, the authors might want to cite papers from the Wildonger lab that suggest mts can be nucleated from Golgi outposts, albeit in a g-tub independent manner.

Rev. 2:

Some of the points previously raised by reviewers have been addressed satisfactorily, however, there are still some concerns, and some points were not adequately addressed (see below). Overall, the manuscript describes some interesting phenotypes and relationships. However, the new interaction experiments performed did not add mechanistic insight into how the players work together, and the final model : Altogether these results argue against kinesin-1 transport over CAMSAP stabilized microtubules and instead suggests that the MTOC vesicles and the CAMSAP puncta are somehow connected" is somewhat unsatisfying- although I am glad that the authors clearly state here that they do not really know what the connection. At minimum, it is essential to better introduce the new protein, Noca-2, at the center of this story (see below). It might be good idea to consult with an evolutionary biologist and map out the relationships in a meaningful way that will provide a good foundation for future work on Noca/ninein proteins.

Major point

The relationship between Noca-1, Noca-2 and ninein is still not clearly described. From the new phylogeny and region diagram in Supplemental Figure 2B and C it looks like Noca-1 and Noca-2 are similar to different regions of the much longer human ninein? The implication is that Noca-1 and Noca-1 have no similarity to one another. It also means that the phylogeny presented in Figure 2B is not done correctly as it does not show evolutionary relationships between similar regions of a protein. My guess based on the information that is presented is that the large ninein protein present in humans is broken into two separate genes in worms. From the information presented it is impossible to tell whether the ancestral form has both regions in one protein, or whether the two pieces were put together in the vertebrate lineage. As a key finding from this paper is the claim that there are two Noca genes in C. elegans, and one is newly described, the relationships between Noca-1, Noca-2 and ninein are essential to get right, and I am not convinced that the current approach is giving an accurate, clear, or informative overview of how the new gene fits into the family.

Minor points

Some of the references are used inaccurately, and may mislead other readers. It will be important to double check that all references actually contain the information they are linked to. Here are some examples:s

I think one of the references in this sentence in the introduction is incorrect: "In mammals, Augmin mediated microtubule nucleation and various MAPs were shown to contribute to the mixed microtubule organization (Cunha-Ferreira et al., 2018; Kapitein and Hoogenraad, 2015; Maniar et al., 2011; Sánchez-Huertas et al., 2016; Yau et al., 2014)." I believe the Maniar paper does not include any mammalian data.

I am also not sure that all of the papers referenced in the following sentence demonstrate microtubule nucleation on organelles in dendrites: "For example, Golgi outposts and early endosomes have been shown to nucleate microtubules in Drosophila dendrites and were suggested to contribute to the minus-end out microtubule organization in these neurites (Mukherjee et al., 2020; Ori-McKenney et al., 2012; Weiner et al., 2020; Ye et al., 2007)." For example, I don't think there is any information on nucleation in the Ye paper. And while the Mukherjee paper has some information about Golgi and potential nucleation in the cell body I do not recall any link between dendritic Golgi and nucleation.

---

## [Editor Report · Decision Letter 3]

3 Sep 2022

Dear Dr Harterink,

Thank you for your patience while we considered your revised manuscript entitled "PTRN-1/CAMSAP and NOCA-2/NINEIN are required for microtubule polarity in Caenorhabditis elegans dendrites" for publication as a Research Article at PLOS Biology. This revised version of your manuscript has been evaluated by the PLOS Biology editors and the Academic Editor.

Based on this assessment, we are likely to accept this manuscript for publication, provided you satisfactorily address the data and other policy-related requests stated below.

In addition, we would like you to define in the abstract all the acronyms used in the title to make it more accessible to a broad audience.

We expect to receive your revised manuscript within two weeks. 

*Published Peer Review History*

*Press*

Sincerely,

Ines

--

Ines Alvarez-Garcia, PhD

Senior Editor

PLOS Biology

DATA POLICY:

Thank you for submiting a file containing the data underlying the graphs shown in the figures. Please also ensure that figure legends in your manuscript include information on WHERE THE UNDERLYING DATA CAN BE FOUND. For example, you can add at the end of each of the corresponding figure legends "The data underlying the graphs shown in the figure can be found in S1_Data."

We require the original, uncropped and minimally adjusted images supporting all blot and gel results reported in an article's figures or Supporting Information files. We will require these files before a manuscript can be accepted so please prepare and upload them now. Please carefully read our guidelines for how to prepare and upload this data: https://journals.plos.org/plosbiology/s/figures#loc-blot-and-gel-reporting-requirements

DATA NOT SHOWN?

- Please note that per journal policy, we do not allow the mention of "data/results not shown", "personal communication", "manuscript in preparation" or other references to data that is not publicly available or contained within this manuscript. Please either remove mention of these data or provide figures presenting the results and the data underlying the figure(s).

---

## [Editor Report · Decision Letter 4]

27 Sep 2022

Dear Dr Harterink,

Thank you for the submission of your revised Research Article entitled "PTRN-1/CAMSAP and NOCA-2/NINEIN are required for microtubule polarity in Caenorhabditis elegans dendrites" for publication in PLOS Biology. On behalf of my colleagues and the Academic Editor, Bing Ye, I am happy to say that we can in principle accept your manuscript for publication, provided you address any remaining formatting and reporting issues. These will be detailed in an email you should receive within 2-3 business days from our colleagues in the journal operations team; no action is required from you until then. Please note that we will not be able to formally accept your manuscript and schedule it for publication until you have completed any requested changes.

PRESS

Sincerely, 

Ines

--

Ines Alvarez-Garcia, PhD

Senior Editor

PLOS Biology
